# FedBiP: Heterogeneous One-Shot Federated Learning with Personalized Latent Diffusion Models

## Abstract

One-Shot Federated Learning (OSFL), a special decentralized machine learning paradigm, has recently gained significant attention. OSFL requires only a single round of client data or model upload, which reduces communication costs and mitigates privacy threats compared to traditional FL. Despite these promising prospects, existing methods face challenges due to client data heterogeneity and limited data quantity when applied to real-world OSFL systems. Recently, Latent Diffusion Models (LDM) have shown remarkable advancements in synthesizing high-quality images through pretraining on large-scale datasets, thereby presenting a potential solution to overcome these issues. However, directly applying pretrained LDM to heterogeneous OSFL results in significant distribution shifts in synthetic data, leading to performance degradation in classification models trained on such data. This issue is particularly pronounced in rare domains, such as medical imaging, which are underrepresented in LDM's pretraining data. To address this challenge, we propose Federated Bi-Level Personalization (FedBiP), which personalizes the pretrained LDM at both instance-level and concept-level. Hereby, FedBiP synthesizes images following the client's local data distribution without compromising the privacy regulations. FedBiP is also the first approach to simultaneously address feature space heterogeneity and client data scarcity in OSFL. Our method is validated through extensive experiments on three OSFL benchmarks with feature space heterogeneity, as well as on challenging medical and satellite image datasets with label heterogeneity. The results demonstrate the effectiveness of FedBiP, which substantially outperforms other OSFL methods.

## 1 Introduction

Federated Learning (FL) (McMahan et al., 2017) is a decentralized machine learning paradigm, in which multiple clients collaboratively train neural networks without centralizing their local data. However, traditional FL frameworks require frequent communication between a server and clients to transmit model weights, which would lead to significant communication overheads (Kairouz et al., 2021). Additionally, such frequent communication increases system susceptibility to privacy threats, as transmitted data can be intercepted by attackers who may then execute membership inference attacks (Lyu et al., 2020). In contrast, a special variant of FL, One-Shot Federated Learning (OSFL) (Guha et al., 2019), serves as a promising solution. OSFL requires only single-round server-client communication, thereby enhancing communication efficiency and significantly reducing the risk of interception by malicious attackers. Therefore, we focus on OSFL given its promising properties.

Despite these promising prospects, existing methods for OSFL face significant challenges when applied to real-world scenarios. Previous works (Guha et al., 2019; Li et al., 2020) require additional public datasets, presenting challenges in privacy-critical domains such as medical data (Liu et al., 2021), where acquiring data that conforms to client-specific distributions is often impractical. Alternatively, they can involve the transmission of entire model weights (Zhang et al., 2022) or local training data (Zhou et al., 2020), which are inefficient and increase the risk of privacy leakage. Moreover, these approaches overlook the issue of feature space heterogeneity, wherein the data features across different clients exhibit non-identically distributed properties. This presents an important and prevalent challenge as emphasized in (Li et al., 2021; Chen et al., 2023). Another vital challenge in

(One-Shot) FL is the limited quantity of data available from clients (McMahan et al., 2017). This problem is particularly notable in specialized domains, such as medical or satellite imaging (So et al., 2022) where data collection is time-consuming and costly.

Data augmentation constitutes a promising strategy to address these challenges in traditional FL (Zhu et al., 2021; Li et al., 2022) by optimizing an auxiliary generative model. However, its reliance on multiple communication rounds makes it unsuitable for OSFL. Recently, diffusion models (Ho et al., 2020), particularly Latent Diffusion Model (LDM) (Rombach et al., 2022), have gained significant attention due to their capability to synthesize high-quality images after being pretrained on large-scale datasets. They are pro-

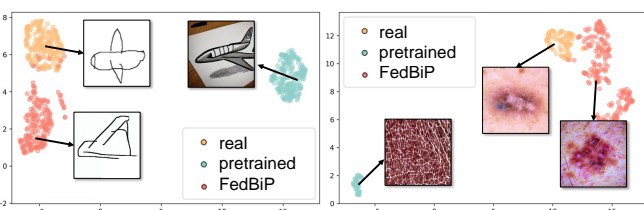

(a) **DomainNet**, airplane, quickdraw    (b) **DermaMNIST**, dermatofibroma class

Figure 1: Feature map visualization of original client images (*real*), synthetic images by prompted pretrained LDM (*pretrained*), and our method (*FedBiP*) on two datasets. `FedBiP` effectively mitigates the strong distribution shifts between pretrained LDM and client local data.

ven effective in various tasks, including training data augmentation (Yuan et al., 2023; Azizi et al., 2023) and addressing feature shift problems (Niemeijer et al., 2024; Gong et al., 2023) under centralized settings. However, directly applying a pretrained LDM for specialized domains presents challenges. As demonstrated in Figure 1, there is a noticeable feature distributional shift and visual discrepancy between real and synthetic data. This mismatch could lead to performance degradation when incorporating such synthetic data into the training process, especially in heterogeneous OSFL settings, where each client possesses data with varying distributions.

Therefore, in this paper, we propose Federated Bi-Level Personalization (`FedBiP`), a framework designed to adapt pretrained LDM for synthesizing high-quality training data that adheres to client-specific data distributions in OSFL. `FedBiP` incorporates personalization of the pretrained LDM at both instance and concept levels. Specifically, instance-level personalization focuses on adapting the pretrained LDM to generate high-fidelity samples that closely align with each client's local data while preserving data privacy. Concurrently, concept-level personalization integrates category and domain-specific concepts from different clients to enhance data generation diversity at the central server. This bi-level personalization approach improves the performance of classification models trained on the synthesized data. Our contributions can be summarized as follows:

- We propose a novel method `FedBiP` to incorporate pretrained Latent Diffusion Model (LDM) for heterogeneous OSFL, marking the first OSFL framework to tackle feature space heterogeneity via personalizing LDM.

- We conduct comprehensive experiments on three OSFL benchmarks with feature space heterogeneity, in which `FedBiP` achieves state-of-the-art results.

- We validate the maturity and scalability of `FedBiP` on real-world medical and satellite image datasets with label space heterogeneity, and further demonstrate its promising capability in preserving client privacy.

## 2 RELATED WORKS

### 2.1 ONE-SHOT FEDERATED LEARNING

A variety of efforts have been made to address One-Shot Federated Learning (OSFL), primarily from two complementary perspectives: one focuses on model aggregation through techniques such as model prediction averaging (Guha et al., 2019), majority voting (Li et al., 2020), conformal prediction method (Humbert et al., 2023), loss surface adaptation (Su et al., 2023), or Bayesian methods (Yurochkin et al., 2019; Chen & Chao, 2020; Hasan et al., 2024). These approaches may not fully exploit the underlying knowledge across different client data distributions. Another aims to transmit training data instead of model weights: data distribution (Kasturi et al., 2020; Beitollahi et al., 2024; Shin et al., 2020), Generative Adversarial Networks (GANs) (Goodfellow et al., 2020; Zhang et al., 2022; Kasturi & Hota, 2023; Kang et al., 2023; Dai et al., 2024), or distilled dataset (Zhou et al.,

2020; Song et al., 2023) are optimized and transmitted to the central server for subsequent model training. Given the success of diffusion models (Rombach et al., 2022), (Zhang et al., 2023; Yang et al., 2024b) suggests transmitting image captions to reproduce training data at the server, while (Yang et al., 2024a) focuses on one-shot semi-supervised FL. However, these approaches are either inefficient or pose risks of client information leakage. In contrast, `FedBiP` functions as an OSFL algorithm, offering enhanced efficiency and robust privacy-preserving capabilities.

## 2.2 DIFFUSION MODELS FOR IMAGE SYNTHESIS

Diffusion models (Ho et al., 2020), especially Latent Diffusion Model (LDM) (Rombach et al., 2022), have attracted significant attention due to their capability to generate high-resolution natural images. They have demonstrated effectiveness in various applications, including image stylization (Guo et al., 2023; Meng et al., 2021; Kawar et al., 2023) and training data generation (Yuan et al., 2023; Sarıyıldız et al., 2023; Azizi et al., 2023). We refer readers to (Croitoru et al., 2023; Yang et al., 2023b) for a comprehensive overview of recent progress on diffusion models. Pretrained LDM has been adopted to address client data scarcity in OSFL (Zhang et al., 2023; Yang et al., 2024b). However, these methods often overlook the feature distribution shift between the LDM pretraining dataset and the clients' local data. This challenge is particularly pronounced in complex domains such as medical and satellite imaging. To address this issue, we propose `FedBiP`, which personalizes the pretrained LDM to synthesize data that is aligned with the clients' data distributions.

## 3 PRELIMINARIES

### 3.1 HETEROGENEOUS ONE-SHOT FEDERATED LEARNING

In this section, we introduce our problem setting, i.e., heterogeneous One-Shot Federated Learning (OSFL). Following (Zhang et al., 2023), we focus on image classification tasks with the goal of optimizing a $C$-way classification model $\phi$ utilizing the client local data, where $C \in \mathbb{N}$ denotes the number of categories. We assume there are $K \in \mathbb{N}$ clients joining the collaborative training. Each client $k$ owns its private dataset $D^k$ containing $N^k \in \mathbb{N}$ (image, label) pairs: $\{x_i^k, y_i^k\}_{i=1}^{N^k}$. Only one-shot data upload from the clients to the central server is allowed.

As described in (Kairouz et al., 2021), OSFL with data heterogeneity is characterized by distribution shifts in local datasets: $P_{\mathcal{XY}}^{k_1} \neq P_{\mathcal{XY}}^{k_2}$ with $k_1 \neq k_2$, where $P_{\mathcal{XY}}^k$ defines the joint distribution of input space $\mathcal{X}$ and label space $\mathcal{Y}$ on $D^k$. Data heterogeneity can be decomposed into two types: (1) *label space* heterogeneity, where $P_\mathcal{Y}$ varies across clients, while $P_{\mathcal{X}|\mathcal{Y}}$ remains the same, and (2) *feature space* heterogeneity, where $P_\mathcal{X}$ or $P_{\mathcal{X}|\mathcal{Y}}$ varies across clients, while $P_{\mathcal{Y}|\mathcal{X}}$ or $P_\mathcal{Y}$ remains the same.

### 3.2 LATENT DIFFUSION MODEL PIPELINE

In this section, we introduce the training and inference pipelines for Latent Diffusion Model (LDM). We provide a schematic illustration in Figure 2. Given an image $x \in \mathbb{R}^{H \times W \times 3}$, the encoder $\mathcal{E}$ encodes $x$ into a latent representation $z(0) = \mathcal{E}(x)$, where $z(0) \in \mathbb{R}^{h \times w \times c}$. Besides, the decoder $\mathcal{D}$ reconstructs the image from the latent, giving $\tilde{x} = \mathcal{D}(z(0)) = \mathcal{D}(\mathcal{E}(x))$. The forward diffusion and denoising processes occur in the latent representation space, as described below.

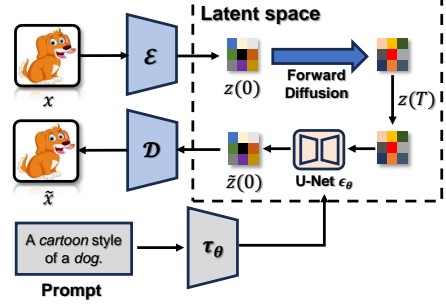

Figure 2: Schematic illustration of the Latent Diffusion Model pipeline with textual prompt conditioning.

In the forward diffusion of LDM training, random noise $\epsilon \sim \mathcal{N}(0, I)$ is added to $z(0)$, producing

$$z(t) = \delta(t, z(0)) = \sqrt{\alpha_t}z(0) + \sqrt{1 - \alpha_t}\epsilon, \quad (1)$$

where $t \sim \text{Uniform}(\{1, ..., T\})$ is the timestep controlling the noise scheduler $\alpha_t$. A larger $t$ corresponds to greater noise intensity. In the denoising process, a UNet $\epsilon_\theta$ is applied to denoise $z(t)$,

Figure 3: Schematic illustration of Federated Bi-Level Personalization (FedBiP). (①) Each client executes bi-level personalization and obtains latent vectors $z^k(T)$ and concept vectors $S^k, V^k$. (②) The central server integrates the vectors into the generation process of the pretrained Latent Diffusion Model $\theta$. (③) The classification model $\phi$ is optimized using synthetic images.

yielding $\tilde{z}(0)$ for image reconstruction. To further condition LDM generation on textual inputs $P$, a feature extractor $\tau_\theta$ is used to encode the prompts into intermediate representations for $\epsilon_\theta$. By sampling different values of $\epsilon$ and $t$, $\epsilon_\theta$ can be optimized via the following loss function:

$$L_{LDM} = \mathbb{E}_{z(0),P,\epsilon,t}\left[||\epsilon - \epsilon_\theta(\delta(t,z(0)),t,\tau_\theta(P))||_2^2\right] \qquad (2)$$

In the inference stage, latent representation $z(T)$ will be sampled directly from $\mathcal{N}(0, I)$, and multiple denoising steps are executed to obtain $\tilde{z}(0)$. The image is then decoded via $\tilde{x} = \mathcal{D}(\tilde{z}(0))$.

# 4 METHODOLOGY

## 4.1 MOTIVATIONAL CASE STUDY

To substantiate the necessity of the proposed method, we present an empirical analysis to address the following research question: *Can pretrained Latent Diffusion Model (LDM) generate images that are infrequently represented in the pretraining dataset using solely textual conditioning?* Specifically, we adopt two datasets, namely DomainNet (Peng et al., 2019) and DermaMNIST (Yang et al., 2023a), which contain images indicating different styles and images from challenging medical domains, respectively. We prompt LDM with *"A quickdraw style of an airplane."* to generate airplane images in quickdraw style for DomainNet dataset, and *"A dermatoscopic image of a dermatofibroma, a type of pigmented skin lesions."* for DermaMNIST. We synthesize 100 images for each setting and adopt a pretrained ResNet-18 (He et al., 2016) to acquire the feature embeddings of real and synthetic images. Finally, we visualize them using UMAP (McInnes et al., 2018).

As shown in Figure 1, we observe markedly different visual characteristics between synthetic and real images. Specifically, for DomainNet, there exist significant discrepancies between the "quick-draw" concept demonstration in the original dataset and the pretrained LDM. For DermaMNIST, the pretrained LDM is only able to perceive the general concepts of "dermatoscopic" and "skin lesion", failing to capture category-specific information. This further highlights the difficulties in reproducing medical domain data via LDM. Additionally, there is a substantial gap in the extracted feature embeddings between real and synthetic images. Most importantly, despite the high visual quality of the synthetic images, they may not contribute to the final performance of the classification model. As demonstrated by our experimental results (Table 5.3), directly applying such prompts to generate images for server-side training sometimes yields worse results than baseline methods. Therefore, it is essential to design a more sophisticated method to effectively personalize the pretrained LDM to the specific domains of client local datasets. These observations motivate our proposed method FedBiP, which mitigates the distribution shifts between pretrained LDM and the client local data. We introduce FedBiP in the following.

## 4.2 PROPOSED METHOD

A schematic overview of the proposed method is provided in Figure 3. Additionally, the pseudo-code of the proposed method is presented in Algorithm 1. We begin by introducing the bi-level personalization in the local update of $k^{th}$ client, omitting the subscript $k$ for simplicity in the following description.

---

**Algorithm 1** Training process of `FedBiP`

---

**ServerUpdate**

1: Initialize Latent Diffusion Model with pretrained weights $\theta$, classification model $\phi$, synthetic dataset $D_{syn} \leftarrow \varnothing$
2: **for** client $k = 1$ to $K$ **do** {**in parallel**}
3:    $k^{th}$ client execute $ClientUpdate(k)$ and upload $\{z_i^k(T), y_i^k\}_{i=1}^{N_k}, \{V_j^k\}_{j=1}^C, S^k$
4:    **for** $i = 1$ to $N_k$ **do**
5:       $e \leftarrow \tau_\theta("A\ [S^k]\ style\ of\ a\ [V_{y_i^k}^k]")$
6:       $\tilde{z}(0) \leftarrow \epsilon_\theta(z_i^k(T), t, e), \tilde{x} \leftarrow \mathcal{D}(\tilde{z}(0))$
7:       $D_{syn}.append([\tilde{x}, y_i^k])$
8: Optimize $\phi$ using $D_{syn}$ (Equation 6)

**ClientUpdate**$(k)$

1: Initialize Latent Diffusion Model with pretrained weights $\theta$, randomly initialize $\{V_j^k\}_{j=1}^C, S^k$.
2: **for** $i = 1$ to $N^k$ **do**
3:    Randomly sample an image $x_{i'}^k$ with $i \neq i', y_i = y_{i'}$
4:    $\overline{z}(0) \leftarrow \gamma \mathcal{E}(x_i^k) + (1 - \gamma)\mathcal{E}(x_{i'}^k)$
5:    $z_i^k(T) \leftarrow \delta(T, \overline{z}(0))$
6: **for** local step $st = 1$ to $N_{step}$ **do**
7:    Sample one mini-batch $\{x_b^k, y_b^k\}$ from $D^k$, timestep $t$
8:    $e \leftarrow \tau(\{"A\ [S^k]\ style\ of\ a\ [V_{y_b^k}^k]"\})$
9:    Optimize $S^k, \{V_j^k\}_{j=1}^C$ (Equation 4)

---

### 4.2.1 INSTANCE-LEVEL PERSONALIZATION

While the traditional Latent Diffusion Model (LDM) employs a Gaussian distribution to initialize the latent vector $z(T) \sim \mathcal{N}(0, I)$, we directly compute $z(T)$ from the local training set $D^k$ of each client. Specifically, we leverage the VAE encoder $\mathcal{E}$ from pretrained LDM to obtain $z_i(T)$ for each specific real sample $x_i$. We first extract the low-dimensional latent representation by feeding the training image into VAE encoder: $z_i(0) \leftarrow \mathcal{E}(x_i)$. We implement additional measures to enhance client privacy. First, we interpolate $z_i(0)$ with another latent representation, $z_{i'}(0)$, from the same class, thereby reducing the risk of exact sample reconstruction. Second, we add $T$-steps of random noise to obtain $z_i(T)$, which corresponds to the maximum noise intensity in LDM. A comprehensive privacy analysis is provided in Section 5.5 and 5.6. The overall process can be formalized as

$$z_i(T) \leftarrow \delta(T, \gamma z_i(0) + (1 - \gamma)z_{i'}(0)), s.t., i \neq i', y_i = y_{i'}, \qquad (3)$$

where $\gamma \sim \mathcal{N}(0.5, 0.1^2)$ and clipped to $[0, 1]$. After the computation, we store $z_i(T)$ and its corresponding ground truth label $y_i$ for all training images in the $k^{th}$ client as the instance-level personalization. We emphasize that this level of personalization does not require any additional optimization, making the process computationally efficient.

### 4.2.2 CONCEPT-LEVEL PERSONALIZATION

Solely applying instance-level personalization results in reduced diversity in image generation. To mitigate this limitation, we enhance personalization by incorporating domain and category concepts into the LDM generation process. Specifically, "domain" denotes the feature distribution within a client's local dataset, such as an image style in the DomainNet dataset. To avoid the costly finetuning of the LDM weights $\theta$, we finetune only the textual guidance. Specifically, we randomly initialize the domain concept vector $S \in \mathbb{R}^{n_s \times d_w}$ and category concept vector $V \in \mathbb{R}^{C \times n_v \times d_w}$, where $n_s$ and $n_v$ are the number of tokens for domain concept and category concept, respectively, and $d_w$ is the token embedding dimension of the textual conditioning model $\tau_\theta$. Subsequently, specific tokens in the textual template $P$ are substituted with the concept vectors $S$ and $V_y$ corresponding to a specific category $y$. For instance, this could result in textual prompts like "A [S] style of a [V_y]" for DomainNet dataset. Following this, $\tau_\theta$ encodes these modified prompts, transforming the textual embeddings into intermediate representation for the denoising UNet $\epsilon_\theta$.

To jointly optimize both concept vectors $S$ and $V_y$, we adopt the following objective function:

Table 1: Evaluation results of different methods on three OSFL benchmarks with feature space heterogeneity. We report the mean±std classification accuracy from 3 runs with different seeds. The best and second-best results are marked with **bold** and underline, respectively.

| Dataset | | FedAvg | Central (*oracle*) | FedD3 | DENSE | FedDEO | FGL | **FedBiP-S** | **FedBiP-M** | **FedBiP-L** |
|---|---|---|---|---|---|---|---|---|---|---|
| Domain Net | C | 73.12 ±1.54 | 73.63 ±0.91 | 61.21 ±1.46 | 63.84 ±2.51 | 72.33 ±1.26 | 67.71 ±3.15 | 68.07 ±0.96 | 74.01 ±1.67 | **77.52** ±0.67 |
| | I | 59.85 ±1.51 | 61.76 ±0.94 | 50.39 ±1.64 | 52.87 ±0.38 | 57.39 ±0.84 | 59.83 ±1.55 | 54.06 ±2.56 | 58.42 ±2.05 | **60.94** ±2.08 |
| | P | 63.77 ±1.12 | 69.18 ±1.74 | 60.50 ±1.09 | 62.07 ±0.97 | 63.17 ±1.05 | **68.56** ±2.51 | 58.24 ±0.22 | 63.01 ±2.25 | 65.20 ±0.78 |
| | Q | 16.26 ±2.60 | 72.83 ±0.82 | 28.25 ±3.11 | 29.92 ±1.62 | 37.86 ±2.47 | 19.83 ±2.99 | 51.09 ±2.05 | 49.64 ±5.05 | **51.85** ±3.24 |
| | R | 87.90 ±0.09 | 87.86 ±0.24 | 79.15 ±1.44 | 81.69 ±1.14 | 81.51 ±1.03 | **87.09** ±0.88 | 80.44 ±1.38 | 82.20 ±0.67 | 83.16 ±0.60 |
| | S | 68.07 ±4.67 | 75.28 ±0.96 | 58.07 ±1.35 | 59.20 ±2.12 | 62.86 ±1.61 | 67.15 ±3.97 | 57.17 ±1.59 | 61.92 ±1.35 | **68.24** ±0.78 |
| | Avg | 61.49 ±0.58 | 73.42 ±0.53 | 56.26 ±0.74 | 58.26 ±1.33 | 62.52 ±1.56 | 61.69 ±1.56 | 61.51 ±0.62 | 64.86 ±0.49 | **67.82** ±0.56 |
| PACS | A | 52.68 ±3.22 | 53.06 ±0.53 | 42.42 ±1.81 | 44.64 ±0.14 | 49.89 ±0.91 | **55.04** ±1.79 | 43.01 ±1.80 | 50.15 ±1.86 | 53.26 ±2.54 |
| | C | 68.27 ±4.22 | 71.43 ±1.61 | 60.47 ±2.46 | 63.10 ±1.47 | 68.31 ±1.41 | 69.94 ±1.43 | 64.58 ±3.23 | 67.71 ±0.93 | **70.90** ±2.97 |
| | P | 86.31 ±1.03 | 81.55 ±6.16 | 72.08 ±2.25 | 74.70 ±0.81 | 71.96 ±0.56 | **76.47** ±0.68 | 70.24 ±2.73 | 73.07 ±1.80 | 74.85 ±1.36 |
| | S | 31.25 ±9.94 | 63.24 ±3.35 | 30.40 ±1.99 | 31.40 ±2.06 | 48.95 ±1.34 | 41.82 ±6.26 | 48.66 ±4.26 | 50.30 ±2.20 | **51.70** ±1.69 |
| | Avg | 59.63 ±3.13 | 67.32 ±2.36 | 51.34 ±2.51 | 53.46 ±1.62 | 59.78 ±1.07 | 60.82 ±1.90 | 56.62 ±1.23 | 60.30 ±0.42 | **62.67** ±0.45 |
| Office Home | A | 54.48 ±1.60 | 58.68 ±1.72 | 50.71 ±1.30 | 52.37 ±0.96 | 49.37 ±2.06 | 48.48 ±3.18 | 39.80 ±0.88 | 45.06 ±0.75 | **55.41** ±0.55 |
| | C | 47.63 ±1.08 | 51.09 ±1.17 | 44.06 ±0.86 | 46.24 ±1.74 | 42.92 ±0.81 | 36.58 ±2.36 | 36.79 ±1.15 | 40.86 ±0.80 | **48.62** ±0.42 |
| | P | 73.94 ±1.27 | 77.79 ±0.83 | 71.09 ±1.69 | 73.76 ±2.07 | 73.81 ±0.46 | 59.38 ±0.66 | 69.20 ±1.17 | 73.23 ±0.69 | **76.63** ±0.20 |
| | R | 63.94 ±0.56 | 69.97 ±0.63 | 60.25 ±0.88 | 61.86 ±1.45 | 61.77 ±0.51 | 62.08 ±2.37 | 56.57 ±1.01 | 61.94 ±1.32 | **65.43** ±0.96 |
| | Avg | 60.00 ±0.88 | 64.38 ±1.06 | 56.52 ±1.07 | 58.55 ±1.35 | 56.96 ±1.71 | 51.63 ±1.71 | 50.59 ±0.70 | 55.27 ±0.73 | **61.52** ±0.39 |

$$L_g = \mathbb{E}_{\mathcal{E}(x(0)),y,\epsilon \sim \mathcal{N}(0,1),t} \left[ ||\epsilon - \epsilon_\theta(z(t), t, \tau_\theta(S, V_y))||_2^2 \right], \tag{4}$$

where timestep $t$ is sampled from $\text{Uniform}(\{1, ..., T\})$.

After the local optimization of each client, the latent vectors $\{z_i(T), y_i\}_{i=1}^{N^k}$, along with the optimized concept vectors $S, V$, are uploaded to the central server. To further increase the generation diversity, we introduce a small perturbation to the domain concept vector $S$. Specifically, we define $\hat{S} = S + \eta$ with $\eta \sim \mathcal{N}(0, \sigma_\eta)$, where $\sigma_\eta$ controls the perturbation intensity. The central server then integrates these vectors into the same pretrained LDM and generates synthetic images with

$$\tilde{x}_i = \mathcal{D}(\epsilon_\theta(z_i(T), T, \tau_\theta(\hat{S}, V_{y_i}))). \tag{5}$$

The data sample $(\tilde{x}_i, y_i)$ is appended to the synthetic set $D_{syn}$. It is crucial to note that FedBiP performs image generation asynchronously, eliminating the need to wait for all clients to complete their local processes. Once the server receives the vectors uploaded from all clients and completes the image generation, we proceed to optimize the target classification model $\phi$ with the objective:

$$L_{cls} = L_{CE}(\phi(\tilde{x}), y). \tag{6}$$

## 5 EXPERIMENTS AND ANALYSES

We conduct extensive empirical analyses to investigate the proposed method. Firstly, we compare FedBiP with other baseline methods on three One-Shot Federated Learning (OSFL) benchmarks with feature space heterogeneity. Next, we evaluate FedBiP using a medical dataset and a satellite image dataset adapted for OSFL setting with label space heterogeneity, illustrating its effectiveness under challenging real-world scenarios. Finally, we perform an ablation study on FedBiP and further analyze its promising privacy-preserving capability.

### 5.1 BENCHMARK EXPERIMENTS

**Datasets Description:** We adapt three common image classification benchmarks with feature distribution shift for our OSFL setting: (1) *DomainNet* (Peng et al., 2019), which contains six domains: Clipart (C), Infograph (I), Painting (P), Quickdraw (Q), Real (R), and Sketch (S). We select 10 categories following (Zhang et al., 2023). (2) *PACS* (Li et al., 2017), which includes images that belong to 7 classes from four domains: Art (A), Cartoon (C), Photo (P), and Sketch (S). (3) *OfficeHome* (Venkateswara et al., 2017) comprises images of daily objects from four domains: Art (A), Clipart (C), Product (P), and Real (R). Each client is assigned a specific domain. To simulate local data scarcity described in previous sections, we adopt 16-shot per class (8-shot for OfficeHome) for each client, following previous works (Li et al., 2021; Chen et al., 2023).

**Baseline Methods:** We compare `FedBiP` with several baseline methods, including *FedAvg* and *Central*, i.e., aggregating the training data from all clients. We note that *Central* is an oracle method as it infringes on privacy requirements, while *FedAvg* requires multi-round communication and is not applicable to OSFL. Besides, we validate concurrent generation-based methods for OSFL: (1) *FedD3* (Song et al., 2023), where distilled instances from the clients are uploaded. (2) *DENSE* (Zhang et al., 2022), where client local models are uploaded and distilled into one model using synthetic images. (3) *FedDEO* (Yang et al., 2024b), where the optimized category descriptions are uploaded and guide pretrained diffusion models. (4) *FGL* (Zhang et al., 2023), where captions of client local images, extracted by BLIP-2 (Li et al., 2023), are uploaded and guide pretrained LDM.

**Implementation Details:** We adopt the HuggingFace open-sourced "CompVis/stable-diffusion-v1-4" as the pretrained Latent Diffusion Model, and use ResNet-18 pretrained on ImageNet (Deng et al., 2009) as the initialization for the classification model. We investigate three variants of `FedBiP`, namely "S", "M", and "L", which corresponds to generating $2\times$, $5\times$, $10\times$ the number of images in the original client local dataset, respectively. Note that synthesizing more images does not affect the client's local optimization costs. We optimize the concept vectors for 50 epochs at each client. For *FGL*, 3500 samples per class per domain are generated. For *FedDEO*, the total number of synthetic images is identical to `FedBiP-L` for a fair comparison. Further details about training hyperparameters are provided in the Appendix.

**Results and Analyses:** We report the validation results in Table 1, where we observe `FedBiP-L` outperforms all baseline methods in average performance, indicating an average performance improvement of up to 5.96%. Notably, `FedBiP-S` achieves comparable performance to *FGL* by generating only 16 images for DomainNet per class and domain, while *FGL* requires 3500 images. This further highlights the efficiency of our proposed method. Additionally, `FedBiP` excels in challenging domains, such as Quickdraw (Q) of DomainNet and Sketch (S) of PACS, showcasing its effectiveness in generating images that are rare in the Latent Diffusion Model (LDM) pretraining dataset. However, `FedBiP` slightly underperforms in certain domains, e.g., Real (R) in Domain-Net. We attribute this to the overlap between these domains and the LDM pretraining dataset, where adapting LDM with the client local datasets reduces its generation diversity. Nevertheless, `FedBiP` narrows the gap between the generation-based methods and oracle `Central` method.

## 5.2 VALIDATION ON MEDICAL AND SATELLITE IMAGE DATASETS

To illustrate the effectiveness of `FedBiP` on challenging real-world applications, we adopt a medical dataset, *DermaMNIST* (Yang et al., 2023a), comprising dermatoscopic images of 7 types of skin lesion, and a satellite image dataset, UC Merced Land Use Dataset (*UCM*) (Yang & Newsam, 2010), which includes satellite images representing 21 different land use categories. We assume there are 5 research institutions (clients) participating in the collaborative training. To construct local datasets for each client in OSFL, we employ the Dirichlet distribution $Dir_\beta$ to model label space heterogeneity, in which a smaller $\beta$ indicates higher data heterogeneity. Following (Zhou et al., 2022), we use the textual template "*A dermatoscopic image of a [CLS], a type of pigmented skin lesions.*" and "*A centered satellite photo of [CLS].*" for DermaMNIST and UCM, respectively.

In Table 2, we report the validation results of different methods on real-world OSFL benchmarks with varying levels of label space heterogeneity. We observe that `FedBiP-L` consistently outperforms all baseline methods across all settings, with an average performance increase of up to 4.16% over *FedAvg*. Furthermore, we notice that the most lightweight version, `FedBiP-S`, surpasses the method with pretrained LDM, *FGL*, by a substantial margin. This demonstrates the importance of

Table 2: Evaluation results of different methods on real-world medical and satellite OSFL benchmarks with varying levels of label space heterogeneity. The best results are marked with **bold**.

| Dataset | Split | FedAvg | Central (*oracle*) | FedD3 | DENSE | FedDEO | FGL | FedBiP-S | FedBiP-M | FedBiP-L |
|---------|-------|--------|--------------------|-------|-------|--------|-----|----------|----------|----------|
| UCM | IID | 63.82 ±0.67 | 68.44 ±0.52 | 59.37 ±1.24 | 64.08 ±0.95 | 63.15 ±0.86 | 52.65 ±1.74 | 61.58 ±0.76 | 63.74 ±0.47 | **65.59** ±1.01 |
| | $Dir_{0.5}$ | 62.96 ±1.41 | 68.44 ±0.52 | 56.86 ±0.81 | 61.41 ±1.51 | 61.04 ±0.34 | 52.65 ±1.74 | 61.02 ±1.03 | 62.37 ±0.84 | **64.41** ±0.88 |
| | $Dir_{0.01}$ | 57.47 ±1.76 | 68.44 ±0.52 | 50.24 ±0.49 | 54.16 ±0.77 | 55.81 ±1.05 | 52.65 ±1.74 | 54.48 ±1.24 | 56.19 ±0.65 | **59.84** ±0.47 |
| Derma MNIST | IID | 53.47 ±1.49 | 60.08 ±0.98 | 50.26 ±0.67 | 52.91 ±0.34 | 54.29 ±1.12 | 40.82 ±2.56 | 53.84 ±1.52 | 54.91 ±0.71 | **56.10** ±1.34 |
| | $Dir_{0.5}$ | 51.98 ±0.52 | 60.08 ±0.98 | 49.52 ±1.46 | 50.83 ±0.61 | 52.61 ±0.84 | 40.82 ±2.56 | 51.47 ±1.32 | 53.26 ±0.84 | **55.03** ±1.02 |
| | $Dir_{0.01}$ | 43.99 ±2.07 | 60.08 ±0.98 | 40.25 ±1.91 | 41.08 ±2.30 | 42.14 ±0.96 | 40.82 ±2.56 | 45.32 ±0.91 | 46.71 ±1.31 | **48.15** ±1.67 |

our LDM personalization schema, particularly in scenarios involving significant feature distribution shifts compared to the pretraining dataset of LDM.

## 5.3 ABLATION STUDY

To illustrate the importance of different `FedBiP` components, we conduct an ablation study on three OSFL benchmark datasets. The results are shown in Table 5.3. First, we observe that simply prompting LDM with *"A [STY] style of a [CLS]"* and synthesizing images at central server is ineffective. Next, we notice that optimizing only the category concept vector $V_c$ leads to only minimal performance improvements. We hypothesize that this is because the categories in these benchmarks are general objects, such as "person" or "clock", which are already well-captured by LDM during pretraining. In contrast, optimiz-

Table 3: Ablation study for different components of `FedBiP` on three benchmarks.

| Instance | Concept | | Domain Net | PACS | Office Home |
|---|---|---|---|---|---|
| $z(T)$ | $\hat{S}$ | $V_c$ | | | |
| FedAvg (*multi-round*) | | | 61.49 | 59.63 | 60.00 |
| | | | 60.22 | 58.90 | 53.23 |
| | | ✓ | 61.71 | 59.15 | 55.81 |
| | ✓ | | 63.96 | 60.08 | 56.32 |
| ✓ | | | 66.08 | 61.83 | 59.35 |
| ✓ | ✓ (no perturb.) | ✓ | 67.09 | 62.78 | 60.84 |
| ✓ | ✓ | ✓ | 67.82 | 62.67 | 61.52 |

ing the domain concept vector $S$ produces visible performance gain. This can be attributed to the mismatch between the textual representation of domain concepts and LDM's pretraining. For example, as described in Motivation section (Figure 1), "Quickdraw" in DomainNet encompasses images characterized by very simple lines, while LDM tends to generate images with finer details. Furthermore, applying instance-level personalization with $z(T)$ yields a performance boost, highlighting the importance of fine-grained personalization in improving LDM. Finally, combining both levels of personalization further improves the results, which demonstrates their complementarity.

## 5.4 SCALABILITY ANALYSIS OF FEDBIP

To show the scalability of `FedBiP` under various application scenarios, we validate `FedBiP` in systems with varying client numbers and analyze the effects of synthetic image quantity.

**Varying Number of Clients:** We split each domain of the DomainNet dataset into 5 subsets, ensuring that each subset contains 16 samples per category to simulate the local data scarcity described in previous sections. Each subset is then assigned to a specific client. In our experiments, we select 1 to 5 clients from each domain, resulting in a total of 6 to 30 clients participating in federated learning.

The validation results are presented in Figure 4. We observe that the performance of the baseline method *FedAvg* remains unchanged with the addition of more clients to FL. In contrast, the validation performance of `FedBiP` consistently increases, narrowing the gap between distributed optimization and *Central* optimization. Furthermore, `FedBiP` outperforms *FedAvg* by $9.51\%$ when the largest number of clients join FL, further indicating its scalability for real-world complex federated systems with more clients.

**Varying Number of Synthetic Images:** We synthesize varying quantities of images for each category and domain, scaling from $1\times$ to $20\times$ the size of the original client local dataset. The results for the DomainNet and OfficeHome benchmarks are presented in Figure 5. Our analysis reveals that increasing the number of synthetic images enhances the performance of the target classification model, significantly outperforming the baseline method (*FedAvg*) by up to $6.47\%$. Furthermore, we

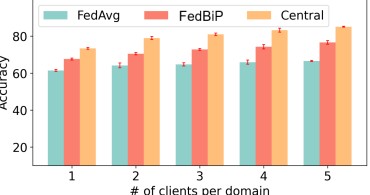

Figure 4: Validation results of `FedBiP` with varying number of clients on DomainNet.

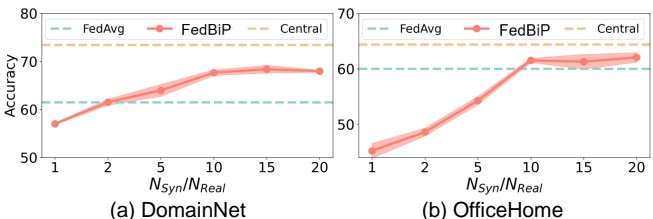

Figure 5: Validation results of `FedBiP` with synthesizing different numbers of images at central server.

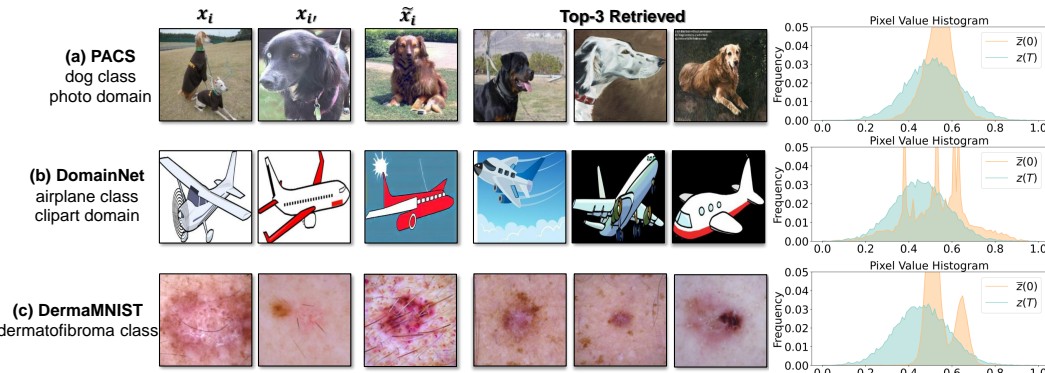

Figure 6: `FedBiP` privacy analysis: (1) **Visual**: The reproduced images are notably dissimilar to the original images $x_i$ and $x_{i'}$. Besides, the retrieved images exhibit visual discrepancies compared to synthetic $\tilde{x}_i$. (2) **Statistical**: The pixel value histogram of $z(T)$ resembles a standard Gaussian distribution more closely compared to $\overline{z}(0)$, making it hard to extract private information from $z(T)$.

observe that synthesizing images at $10\times$ the original dataset size emerges as the most effective approach, when considering the trade-off between generation time and final performance. This finding is consistent with the design principles of `FedMLA-L`.

## 5.5 PRIVACY ANALYSIS

In this section, we present a comprehensive privacy analysis of `FedBiP`, encompassing both qualitative and quantitative evaluations, as illustrated in Figure 6.

**Visual discrepancy between synthetic and real images**: We visualize both synthetic image $\tilde{x}_i$, and its corresponding real images, i.e., $x_i$, $x_{i'}$. Besides, we use the pretrained ResNet-18 to extract the feature map of $\tilde{x}_i$ and retrieve the top-3 real images which indicate the largest cosine similarities in the feature space. We observe differences in both background (e.g., textual and color) and foreground (e.g., the exact object shape, position, and pose) between real and synthetic images. These visual discrepancies indicate that the synthetic images do not closely resemble any individual real images, thereby reducing the risk of revealing sensitive information about the original client data.

**Pixel Value Histogram Analysis**: To further analyze `FedBiP` from a statistical perspective, we provide histograms of both $\overline{z}(0)$ (the interpolated latent vectors of input images) and the corresponding $z(T)$ ($\overline{z}(0)$ with $T$-steps of random noise added). We observe that $z(T)$ closely resembles a standard Gaussian distribution, which contains less information about the original input images compared to $\overline{z}(0)$. This indicates that transmitting the noised $z(T)$ is more private than $\overline{z}(0)$, and would not significantly compromise privacy regulations. Additionally, we notice that $\overline{z}(0)$ could be further replaced with the average latent vectors of all samples from a specific class, i.e., categorical prototypes (Tan et al., 2022). This substitution might further protect client privacy and is appropriate for applications with stringent privacy requirements. We leave this for future work.

**Membership Inference Attack (MIA) Analysis**: Finally, we analyze the resilience of `FedBiP` against MIA. Following (Yeom et al., 2018; Salem et al., 2018), we compute the average loss and entropy of the final model on both training member and non-member data, and report the difference between the two averages. A smaller difference corresponds to better membership privacy preservation. From the MIA Analysis in Table 4, we can observe that `FedBiP` demonstrates superior resilience against MIA.

Table 4: Membership Inference Attack (MIA) analysis on different benchmarks. A lower metric corresponds to better MIA privacy.

| Dataset | MIA Metric | FedAvg | **FedBiP** |
|---|---|---|---|
| DomainNet | Entropy ↓ | 0.1311 | 0.0186 ↓**85.8%** |
| | Loss ↓ | 0.5976 | 0.1611 ↓**73.0%** |
| DermaMNIST | Entropy ↓ | 0.0897 | 0.0551 ↓**38.6%** |
| | Loss ↓ | 0.5860 | 0.4127 ↓**29.6%** |
| PACS | Entropy ↓ | 0.1635 | 0.0338 ↓**79.3%** |
| | Loss ↓ | 0.4459 | 0.1244 ↓**72.1%** |

## 5.6 VISUALIZATION WITH VARYING $\gamma$

In this section, we visualize the synthetic image $\tilde{x}_i$ using different interpolation coefficients $\gamma$ for DomainNet benchmark. Specifically, we compute the interpolated latent vector $\bar{z}_i(0)$ using $\gamma z_i(0) + (1 - \gamma)z_{i'}(0)$. As shown in Figure 7, we observe that the synthetic images exhibit distinct visual characteristics compared to the real images, even when $\gamma$ is set to 0.0 or 1.0, corresponding to the direct use of latent vectors from the original images. We attribute these differences to the sampling process involved in the denoising phase of Latent Diffusion Model. Additionally, applying $\gamma$ values near 0.5 offers the most effective privacy protection. Most importantly, varying $\gamma$ produces diverse images, which enhances generation diversity and is beneficial for training the classification model. Therefore, we use a Gaussian distribution $\mathcal{N}(0.5, 0.1^2)$ to sample $\gamma$ in FedBiP.

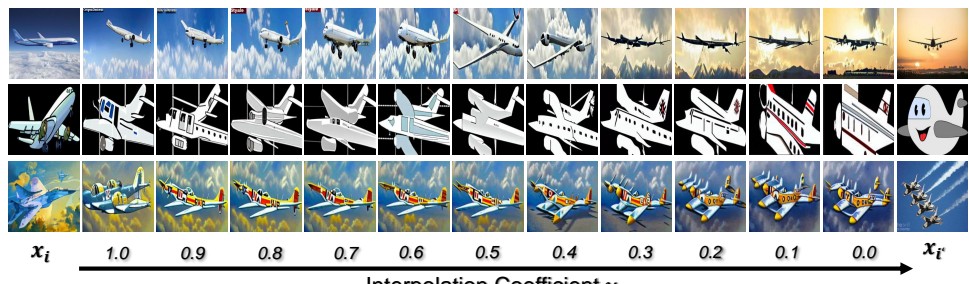

Figure 7: Synthetic images generated with varying $\gamma$ for latent embedding interpolation.

## 5.7 VISUALIZATION FOR CHALLENGING DOMAINS

In this section, we present the synthetic images generated for the challenging domains, i.e., Quickdraw (DomainNet) and Sketch (PACS), as shown in Figure 8. Our observations indicate that FedBiP achieves superior generation quality by more accurately adhering to the original distribution of clients' local data compared to the diffusion-based method FGL (Zhang et al., 2023). This visualization further highlights the effectiveness of our bi-level personalization approach.

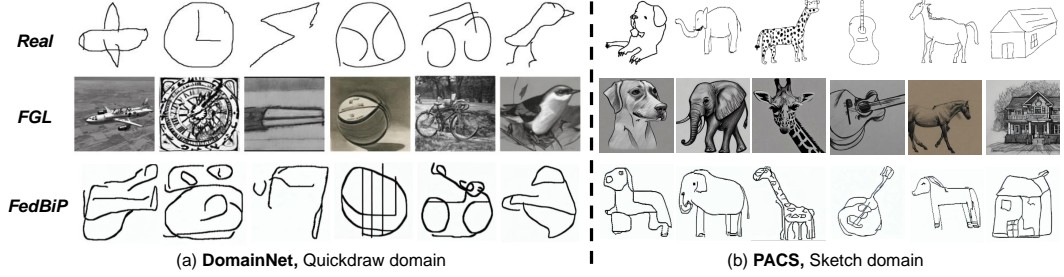

(a) **DomainNet,** Quickdraw domain     (b) **PACS,** Sketch domain

Figure 8: Comparison of synthetic images for challenging domains.

## 6 CONCLUSION

In this work, we propose the first framework to address feature space heterogeneity in One-Shot Federated Learning (OSFL) using generative foundation models, specifically Latent Diffusion Model (LDM). The proposed method, named FedBiP, personalizes the pretrained LDM at both instance-level and concept-level. This design enables LDM to synthesize images that adhere to the local data distribution of each client, exhibiting significant deviations compared to its pretraining dataset. The experimental results indicate its effectiveness under OSFL systems with both feature and label space heterogeneity, surpassing the baseline and multiple concurrent methods. Additional experiments with medical or satellite images demonstrate its maturity for challenging real-world applications. Moreover, additional analysis highlights its promising scalability and privacy-preserving capability.

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

Table 5: Detailed hyperparameters for each dataset. The highlighted words ([STY]) in the textual prompt will be replaced by the domain concept vectors. The [CLS] will be replaced by the class concept vectors.

| Dataset | prompt | $n_s$ | $n_c$ | $C$ | Class Names |
|---|---|---|---|---|---|
| Derma MNIST | A dermatoscopic image of a [CLS], a type of pigmented skin lesions. | 2 | 4 | 10 | intraepithelial carcinoma, basal cell carcinoma, benign keratosis, dermatofibroma, melanoma, melanocytic nevi, vascular skin |
| UCM | A centered satellite photo of [CLS]. | 3 | 3 | 21 | agricultural, dense residential, medium residential, sparse residential, parking lot, buildings, harbor, mobile homepark, storage tanks, freeway, intersection, overpass, golf course, baseball diamond, runway, tenniscourt, beach, forest, river, chaparral, airplane |
| Domain Net | A [STY] of [CLS]. | 1 | 1 | 10 | airplane, clock, axe, basketball, bicycle, bird, strawberry, flower, pizza, bracelet |
| Office Home | A [STY] of [CLS]. | 1 | 1 | 20 | Marker, Spoon, Pencil, Speaker, Toys, Fan, Hammer, Notebook, Telephone, Sink, Chair, Fork, Kettle, Bucket, Knives, Monitor, Mop, Oven, Pen, Couch |
| PACS | A [STY] of [CLS]. | 1 | 1 | 7 | dog, elephant, giraffe, guitar, horse, house, person |

## A EXPERIMENTAL DETAILS

We use 1 NVIDIA RTX A5000 with 24GB RAM to run the experiments. We use PyTorch (Paszke et al., 2019) to implement our algorithm. For the baseline FedAvg, the total communication round is set to 50. For FGL (Zhang et al., 2023), we generate 3500 images per class per domain. For the optimization of the classification model, we use SGD with momentum as the optimizer, where the learning rate is set to 0.01 and the momentum is 0.9. The optimization epoch is set to 50. The training image resolution is set to $512 \times 512$ for all datasets.

For FedD3 (Song et al., 2023), we adopt Kernel Inducing Points (KIP) to distill the original dataset into 1 image per class per domain and transmit them to the central server. For DENSE (Zhang et al., 2022), we first finetune the pretrained ResNet-18 (He et al., 2016) at each client and then optimize a Generator to conduct model distillation at central server. The hyperparameters used in these methods are following their original papers. For `FedMLA`, we use Adam optimizer to optimize the concept vectors. The learning rate is set to 0.1 and beta is set to (0.9, 0.999). The total training epochs is set to 30. We adopt the Pseudo Numerical Diffusion Model (PNDM) (Liu et al., 2022) in the Latent Diffusion Model. The perturbation intensity for domain concept vector $\sigma_\mu$ is set to 0.1 for all dataset. More dataset specific hyperparameters are provided in Table 5.

## B   SYNTHETIC IMAGE VISUALIZATION

We provide synthetic images for all benchmarks in the following figures, where we observe that the synthetic images generally follow the distribution and characteristics of the original training datasets at each client. Besides, the visual quality of the generated images, e.g., the detailed features of the objects, is also promising.

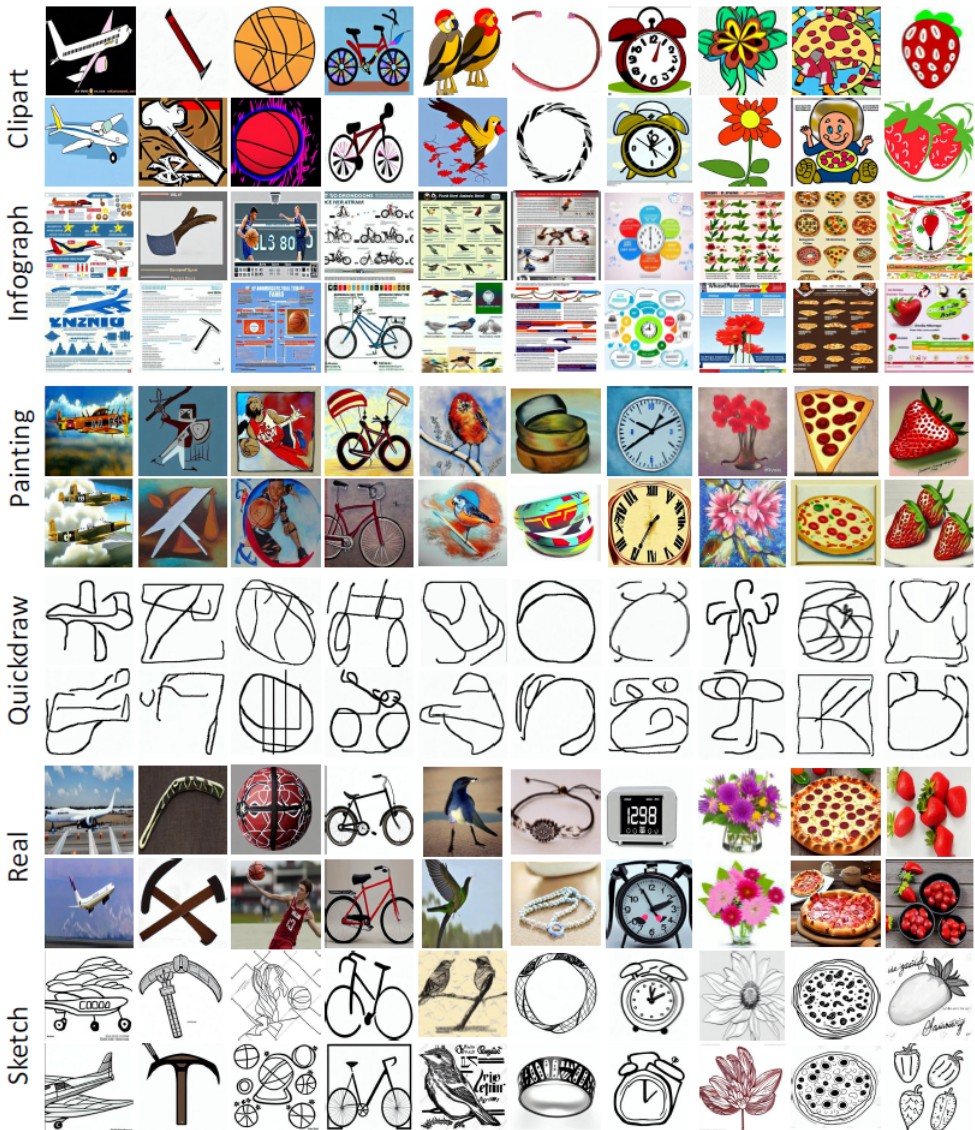

Figure 9: Synthetic Images for DomainNet benchmark.

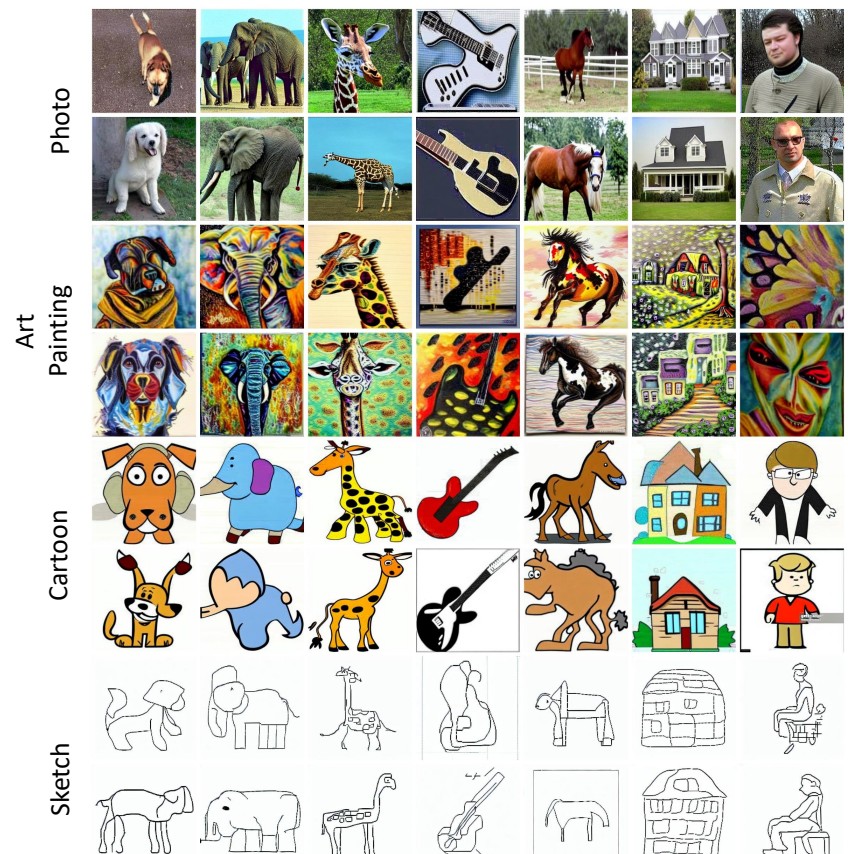

Figure 10: Synthetic Images for PACS benchmark.

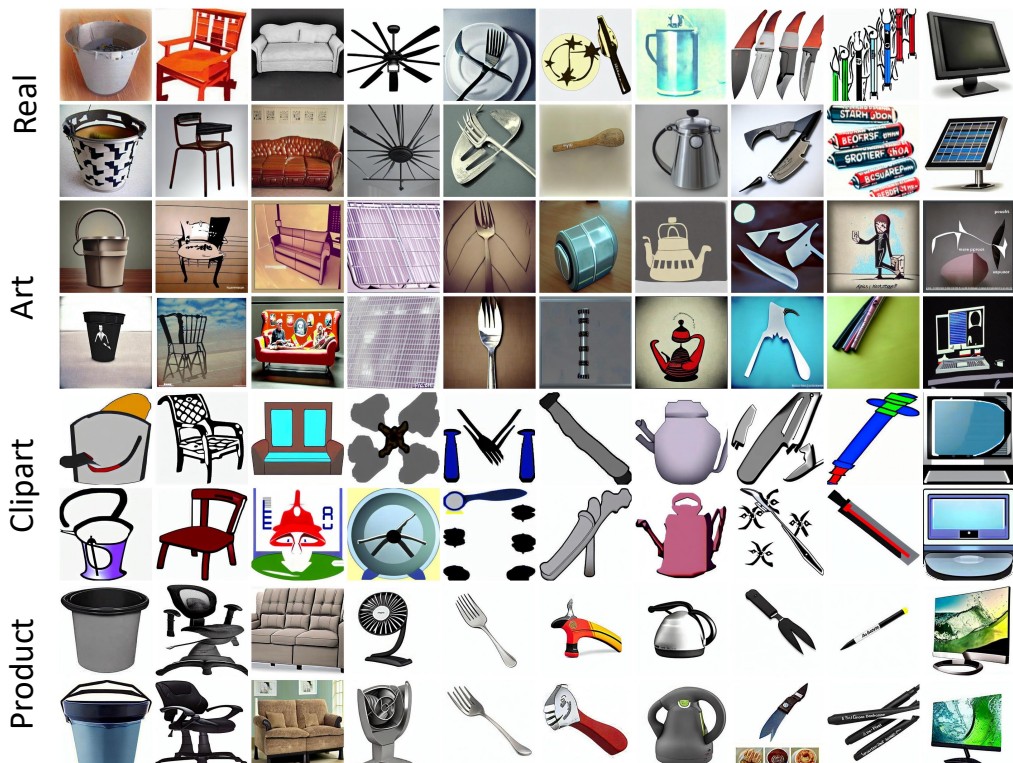

Figure 11: Synthetic Images for OfficeHome benchmark.

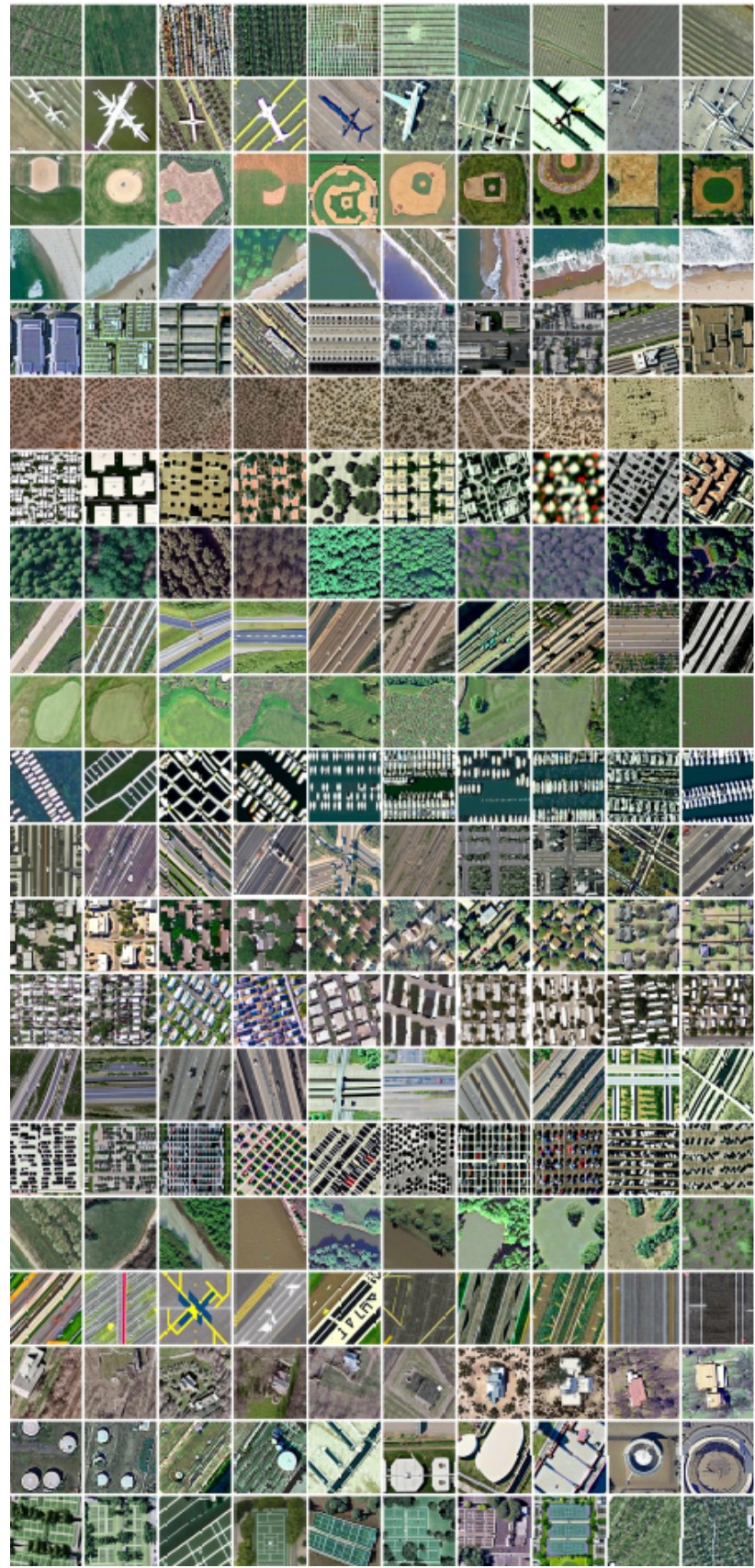

Figure 12: Synthetic Images for UCM benchmark.

Figure 13: Synthetic Images for DermaMNIST benchmark.

