# OpenReview forum: "FedBiP: Heterogeneous One-Shot Federated Learning with Personalized Latent Diffusion Models"
_ICLR.cc/2025/Conference — ICLR 2025 Conference Withdrawn Submission_

### Official Review · Reviewer_uSVt · 2024-10-29

**Soundness:** 3
**Presentation:** 3
**Contribution:** 2
**Rating:** 3
**Confidence:** 5

**Summary:**

This work proposed heterogeneous one-shot federated learning using an improved diffusion-based data agumentation, which can reduce the distribution gap between simulated heterogeneous data and real-world data. Extensive experiments demonstrate that the proposed model significantly outperforms the baseline in heterogeneous federated learning.

**Strengths:**

- Developed a latent-diffusion-model-based data augmentation method to address the issue of insufficient data in heterogeneous federated learning.
- Conduct extensive experiments to show the proposed method performs significantly better than the baseline.

**Weaknesses:**

- FedBiP requires uploading latent features to the server, which could potentially lead to data reconstruction attack. Please provide a data reconstruction attack privacy analysis.
- LDMs are typically large and computationally intensive. Performing bi-level personalization on client devices may impose significant computational and storage burdens, particularly on resource-constrained devices like mobile phones or edge devices. Please provide the time complexity and space complexity analysis in the inference process.
- Personalizing the model through instance-level and concept-level tuning increases system complexity. This added complexity might pose challenges in management and implementation. Discussing ways to reduce the computation and storage requirements on the client side can enhance the quality of this manuscript.
- The performance of LDMs is highly dependent on the quality and relevance of their pretraining data. If the pretraining data does not sufficiently represent the target domain, issues related to distribution shift may arise, potentially degrading the performance of classification models trained on synthetic data. It would be valuable for the authors to discuss how FedBiP might perform under significant domain shifts and explore potential mitigation strategies, such as domain adaptation or fine-tuning with domain-specific data.
- Although the study mentions that FedBiP outperforms existing methods, further validation under different experimental settings, such as imbalanced samples, and on larger-scale datasets, such as ImageNet, CIFAR-10, or CIFAR-100, may still be needed to confirm its effectiveness and scalability.
- The use of LDMs for data augmentation has been widely discussed in the community [1][2][3]. The core contribution of this manuscript lies in proposing a bi-level strategy to improve this augmentation approach. However, the quantitative results show only limited improvements, while the method introduces additional computational overhead on the client side. Moreover, uploading latent space features raises the risk of data reconstruction attacks, which should be carefully considered.

[1] Morafah, M., Reisser, M., Lin, B., & Louizos, C. (2024). Stable Diffusion-based Data Augmentation for Federated Learning with Non-IID Data. arXiv preprint arXiv:2405.07925.
[2] Yang, M., Su, S., Li, B., & Xue, X. (2024, March). Exploring One-Shot Semi-supervised Federated Learning with Pre-trained Diffusion Models. In Proceedings of the AAAI Conference on Artificial Intelligence (Vol. 38, No. 15, pp. 16325-16333).
[3] Yang, M., Su, S., Li, B., & Xue, X. (2024). FedDEO: Description-Enhanced One-Shot Federated Learning with Diffusion Models. arXiv preprint arXiv:2407.19953.

**Questions:**

- Is the performance improvement attributed to the prior knowledge introduced by LDMs?
- What is the novelty of this work compared to other methods that use LDMs for data augmentation in federated learning?

---

> ### Author Response · Authors · 2024-11-14
> **Rebuttal for Reviewer uSVt**
>
> We thank the reviewer for their valuable insights and address each point in detail below:
>
> W1: We kindly refer the reviewer to our experimental results in Figure 6 and Figure 7, where we conduct reconstruction attacks.
> W2/W3: We will include additional analysis on the computational costs of FedBiP in the revised version.
> W4: We refer the reviewer to our experiments on satellite images and medical images, which differ from the pretraining data of the latent diffusion models. Our results show that FedBiP outperforms other methods.
> W5: We appreciate the reviewer’s suggestion. However, we assume that both ImageNet and CIFAR10 datasets contain natural images, which are included in the pretraining of the latent diffusion models, so we have not considered them in our experiments.
> W6: We thank the reviewer for the suggestion. However, we cannot compare with [1] as their method is not applicable for OSFL, and [2] focuses on semi-supervised FL, which is also outside the scope of the current paper. We have compared our approach with [3], i.e., FedDEO, in our main paper (Tables 1 and 2).
> Q1: We assume that the pretraining of latent diffusion models (LDMs) is essential for achieving performance improvements.
> Q2: We are the first to personalize LDMs and demonstrate the effectiveness of FedBiP with images that are under-represented in the pretraining data.

---

### Official Review · Reviewer_HP3z · 2024-11-02

**Soundness:** 3
**Presentation:** 3
**Contribution:** 3
**Rating:** 6
**Confidence:** 4

**Summary:**

The paper proposes a novel method called FedBiP, which incorporates a pretrained Latent Diffusion Model (LDM) for heterogeneous one-shot federated learning (OSFL). This marks the first OSFL framework designed to address feature space heterogeneity through the personalization of LDM. The authors conduct comprehensive experiments on three OSFL benchmarks characterized by feature space heterogeneity, demonstrating that FedBiP achieves state-of-the-art results. Additionally, the maturity and scalability of FedBiP are validated on real-world medical and satellite image datasets featuring label space heterogeneity, highlighting its promising capabilities in preserving client privacy.

**Strengths:**

1.  The integration of a pretrained LDM into the OSFL framework represents a significant advancement in addressing feature space heterogeneity, showcasing creativity and depth in the methodology.
2.  The extensive experiments conducted across various benchmarks effectively demonstrate the robustness and effectiveness of FedBiP, reinforcing its potential impact in the field.
3.  Validating the method on real-world datasets, particularly in sensitive domains like medical imaging, underscores the practical applicability and relevance of the proposed approach.

**Weaknesses:**

1.  The paper lacks a thorough analysis of the time consumption and communication costs associated with the FedBiP method. Understanding these aspects is crucial, particularly in federated learning settings where resource constraints are common. An evaluation of the efficiency of the model updates and the overhead introduced by the personalized LDM would provide valuable insights.
2.  While the use of LDM for generating samples may enhance data privacy, there is a potential risk that the generated samples could be too similar to the original dataset. This similarity could inadvertently expose sensitive information about the clients’ data, raising privacy concerns. A discussion on how to mitigate these risks and ensure that the generated samples maintain sufficient divergence from the original data would be beneficial.

**Questions:**

1.  What are the time consumption and communication costs associated with FedBiP, particularly when scaling to larger datasets or more clients? Providing insights or metrics on these aspects would help evaluate the practical applicability of your method in real-world scenarios.
2.  Given that the samples generated by the LDM may resemble the original datasets, what measures are in place to ensure that client privacy is preserved? Could you elaborate on how you mitigate the risk of sensitive information being inadvertently exposed through these generated samples?

---

> ### Author Response · Authors · 2024-11-14
> **Rebuttal for Reviewer HP3z**
>
> We thank the reviewer for their valuable insights and address each point in detail below:
>
> W1/Q1: We will include additional analysis on the communication costs of FedBiP in the revised version.
> W2/Q2: We refer the reviewer to our privacy analysis in the main paper, where we evaluate FedBiP against multiple attacks, e.g., membership inference attacks (MIA), reconstruction attacks. Our results show that the synthetic images do not resemble the original dataset.

---

### Official Review · Reviewer_w57F · 2024-11-03

**Soundness:** 3
**Presentation:** 3
**Contribution:** 2
**Rating:** 3
**Confidence:** 4

**Summary:**

This paper studies one-shot federated learning  (OSFL) and aims to address the data heterogeneity and limited data quantity issue. A personalized version of the latent diffusion model is proposed to address these issues and the proposed method is evaluated on five public datasets with improved performance over compared methods.

**Strengths:**

- Data heterogeneity and limited data quantity are important topics in FL.
- Using the latent diffusion model to address the data quantity issue is promising.
- Evaluations are performed on multiple different datasets.

**Weaknesses:**

- The motivation for studying OSFL needs to be further justified. It takes great effort to build a FL collaboration, but only one-shot communication is performed. This does not make sense in real-world scenarios, as the FL collaboration efforts have not been fully utilized. Furthermore, privacy threats could be defined by using privacy protection methods such as secure multi-party computation, homomorphic encryption, differential privacy, etc. Performing one-shot communication may not be the ideal solution.
- It is not clear which part needs to be communicated and which parts are preserved locally. It seems only the latent vectors will be uploaded to the server.
- Finetuning the LDM on local client data should be a straightforward solution, which needs to be discussed and compared in experiments.
- It may not be proper to claim the application on the medical dataset as a real-world application, the DermaMNIST has a very low resolution of images, while in the medical area, the image size could be large.

**Questions:**

- With more synthesized images, the performance seems saturated, what could be the reason?
- Why don’t consider the segmentation task?

---

> ### Author Response · Authors · 2024-11-14
> **Rebuttal for Reviewer w57F**
>
> We thank the reviewer for their valuable insights and address each point in detail below:
>
> W1: We kindly refer the reviewer to [1] for the motivation behind OSFL.
> W2: The latent vectors will be uploaded to the central server.
> W3: In our experimental setups, we assume that the local client lacks sufficient computational power for full-sized model finetuning.
> W4: The experiments on DermaMNIST aims at indicating the effectiveness of FedBiP on datasets that differ from the pretraining data of the latent diffusion models.
> Q1: We assume that even though the synthetic images are different, their feature embeddings do not contribute to the classification model performance after a specific threshold.
> Q2: We thank the reviewer for the suggestion and will consider adding more tasks in the revised version.
>
> [1] Li, Qinbin, Bingsheng He, and Dawn Song. "Practical one-shot federated learning for cross-silo setting." arXiv preprint arXiv:2010.01017 (2020).

---

### Official Review · Reviewer_qS6C · 2024-11-03

**Soundness:** 4
**Presentation:** 4
**Contribution:** 4
**Rating:** 5
**Confidence:** 4

**Summary:**

This paper discusses One-Shot Federated Learning (OSFL), a decentralized machine learning approach that minimizes communication costs and enhances privacy by requiring only a single round of client data or model upload. Existing methods encounter challenges related to client data heterogeneity and limited data availability, particularly when applied in real-world contexts. The authors highlight the advancements of Latent Diffusion Models (LDM) in synthesizing high-quality images from large-scale datasets. Despite this potential, directly applying pretrained LDM in heterogeneous OSFL leads to distribution shifts in synthetic data, resulting in degraded performance for classification models, especially in rare domains like medical imaging.

To tackle these issues, the authors introduce Federated Bi-Level Personalization (FedBiP), which personalizes pretrained LDMs at both the instance and concept levels. The effectiveness of FedBiP is demonstrated through extensive experiments on three OSFL benchmarks and challenging datasets in medical and satellite imaging.

**Strengths:**

This paper addresses an existing problem with innovative approaches. The originality is solid, and the topic is significant.

**Weaknesses:**

Limited Discussion of Limitations in Prior Work

1. Although the authors mention FedD3, DENSE, FedDEO, and FGL in their paper, they do not thoroughly examine the technical innovations in comparison to these works. Notably, the paper at this link (https://arxiv.org/html/2306.16064v2) also utilizes generative models for federated learning. What are the key differences? Please provide a comparison with these existing state-of-the-art methods from a methodological perspective. What distinguishes the proposed method as superior to the others?

2. For instance, the authors mention that these approaches are either inefficient or pose risks of client information leakage, but it is unclear how these methods are inefficient or what specific risks they present.

Method Details

1. The authors assert that the concepts are initialized randomly. Do they need to label the datasets to define these concepts, or are the concepts learned by the network itself? How do they ensure that the concepts are sufficiently accurate? If the concepts are incorrect, how might this affect the results? Please provide more details on the initialization and learning process of the concepts, and discuss the potential impacts on results if the concepts are inaccurate.

2. What do FedBiP-S, FedBiP-M, and FedBiP-L represent? How do their parameters compare to those of other methods? Do the authors utilize more parameters than other approaches?

3. Will the generated synthetic images need to be synthesized again during training, or will this be completed beforehand?

4. In Table 3, what experiments are represented in Row 2, the one adjacent to FedAVG?

Rationale

5. It appears that the proposed method could be applicable to other tasks, such as using diffusion models to address limited data problems in image classification under domain shifts. Why do the authors not demonstrate their approach on general benchmarks? What is the specific relationship of the proposed method to federated learning? It does not seem to address the unique challenges inherent in FL. Please discuss potential extensions to other tasks or domains, and to more explicitly connect their approach to specific federated learning challenges.

**Questions:**

Please see my comments above. I will increase my score if the authors could address my concerns well.

---

> ### Author Response · Authors · 2024-11-14
> **Rebuttal for Reviewer qS6C**
>
> We thank the reviewer for their valuable insights and address each point in detail below:
>
> Limited Discussion of Limitations in Prior Work:
> The paper mentioned by the Reviewer (https://arxiv.org/html/2306.16064v2) discusses the FGL approach, which we have compared in our main paper. We outline the primary distinctions as follows:
> (1) FedD3: This approach transmits distilled client datasets to the central server, which poses a risk of revealing information about the clients' local datasets.
> (2) DENSE: This method optimizes generative models on each client, thereby introducing additional computational overhead at the client level.
> (3) FedDEO: This method does not align latent diffusion models with the data distributions of client datasets, which reduces its effectiveness.
> (4) FGL: This approach requires generating 2k images per class at the central server, which is not efficient.
> In contrast, FedBiP is the first to personalize LDMs and effectively handle images that are under-represented in the pretraining data. We will include this expanded discussion in the revised version.
>
> Method Details:
> 1. FedBiP does not require initialization with category labels. We will conduct an additional ablation study to highlight the performance differences with and without category label initialization.
> 2. FedBiP-S/M/L represents generating 2, 5, 10 times of the client local images at the central server. We do not introduce additional parameters compared to the existing methods.
> 3. Image synthesis occurs prior to optimizing the classification model.
> 4. This approach leverages pretrained latent diffusion models to generate synthetic training images.
>
> Rationale:
> We appreciate the reviewer’s insightful comment. Our focus on one-shot federated learning (OSFL) stems from our belief that this approach presents a promising solution for federated learning scenarios, particularly when compared to general benchmark applications. The bi-level personalization framework enables effective results with minimal image generation, making it well-suited for computationally critical FL applications.

---

### Note · Authors · 2024-11-14

**Comment:**

We thank the reviewer for their valuable feedback and would like to withdraw our submission.

**Withdrawal Confirmation:**

I have read and agree with the venue's withdrawal policy on behalf of myself and my co-authors.